# Grain Characteristics, Moisture, and Specific Peptides Produced by *Ustilaginoidea virens* Contribute to False Smut Disease in Rice (*Oryza sativa* L.)

**DOI:** 10.3390/biom13040669

**Published:** 2023-04-12

**Authors:** Robinson C. Jose, Thangjam Kanchal, Bengyella Louis, Narayan C. Talukdar, Devasish Chowdhury

**Affiliations:** 1Institute of Advanced Study in Science and Technology, Guwahati 781035, India; 2Institute of Bioresources and Sustainable Development (IBSD), Imphal 795001, India; 3Department of Plant Sciences, University Park, Pennsylvania State University, 101 Tyson Bldg, State College, PA 16802, USA; 4Faculty of Science, Assam Down Town University, Guwahati 781026, India

**Keywords:** phytopathology, microscopy, proteome, SDS-PAGE, LC-MS/MS

## Abstract

The fungus *Ustilaginoidea virens,* the causative agent of false smut in rice (*Oryza sativa* L.), is responsible for one of the severe grain diseases that lead to significant losses worldwide. In this research, microscopic and proteomic analyses were performed by comparing *U. virens* infected and non-infected grains of the susceptible and resistant rice varieties to provide insights into the molecular and ultrastructural factors involved in false smut formation. Prominent differentially expressed peptide bands and spots were detected due to false smut formation as revealed by sodium dodecyl-sulfate polyacrylamide gel electrophoresis (SDS-PAGE) and two-dimensional gel electrophoresis (2-DE) SDS-PAGE profiles and were identified using liquid chromatography-mass spectrometry (LC-MS/MS). The proteins identified from the resistant grains were involved in diverse biological processes such as cell redox homeostasis, energy, stress tolerance, enzymatic activities, and metabolic pathways. It was found that *U. virens* produces diverse degrading enzymes such as β-1, 3-endoglucanase, subtilisin-like protease, putative nuclease S1, transaldolase, putative palmitoyl-protein thioesterase, adenosine kinase, and DNase 1 that could discretely alter the host morphophysiology resulting in false smut. The fungus also produced superoxide dismutase, small secreted proteins, and peroxidases during the smut formation. This study revealed that the dimension of rice grain spikes, their elemental composition, moisture content, and the specific peptides produced by the grains and the fungi *U. virens* play a vital role in the formation of false smut.

## 1. Introduction

The interaction between plants and their fungal counterpart is a complex process involving specific adaptations between the interacting partners. Since these associations are sometimes harmful to plants, they need to be controlled to improve crop health. Rice is one of the world’s most essential foodstuffs, especially for the Asian population, where it acts as the staple food for more than 1.3 billion people. Among all the regions of the world, South and East Asia are the two major rice-producing regions, with China being the most-producing country with 147 million metric tons (MMT), followed by India with 124 MMT. It has been estimated by the United States Department of Agriculture (USDA) that the production of rice for 2022/2023 will be 503.27 MMT, compared to 515.05 MMT of last year, a decrease of 2.29% [1].

False smut disease of rice affects rice panicles and is known to cause about 0.2–49% yield loss in the various States of India depending on the disease severity and rice variety [2,3,4,5,6,7] and has been reported worldwide [8,9,10,11]. The fungus *Ustilaginoidea virens* is a pathogen of rice shown to cause false smut in the early flowering stage [12]. It has been suggested that signaling cues for *U. virens* infection are the rich nutrients in rice grain [13,14]. Often, *U. virens* grow out of rice spikelets and form white balls, which change in color over time to yellow, orange, and green, and finally to greenish-black smut balls [15].

Regarding the *U. virens* life cycle, doubts remain about whether sclerotia or chlamydospores are the primary inoculums in the field [15]. Previously, it was concluded that sexual and asexual stages are involved in the life cycle of *U. virens* [16]. In the sexual cycle, sclerotia appear on the rice false smut balls [17]. The sclerotia, under suitable conditions, germinate and differentiate into the stroma and then develop into asci with ascospores [18]. The secondary conidia are produced from the ascospores, contributing to rice’s primary infections [19]. While in the asexual cycle, thick-walled chlamydospores form on the surfaces of false smut balls and serve as an essential inoculum source [20]. On the other hand, it is suggested that ascospores and sclerotia could overwinter and cause infection of the host [19,21,22].

Consistent rice production is key to food security for the majority of the human population and also for economic, social, and political stability. The fungus is evolving and bothering rice production by infecting different rice varieties all over the world, which needs to be addressed by understanding the fungus and the host better.

To the best of our knowledge, there is no explanation for why some grains are resistant to the fungus *U. virens*. It is crucial to understand the mechanism of rice infection by *U. virens* to design effective and sustainable management strategies for the false smut disease. In this regard, the first objective of this study was to determine the microstructural interaction of *U. virens* with different rice varieties to generate more insights into this infection process under natural conditions. The second goal of this study was to understand the proteome changes *U. virens* brings in the interactome of the infected rice grains compared to the healthy grains.

## 2. Materials and Methods

### 2.1. Survey and Sample Collection

The rice samples were observed and collected from two States of Northeast India (NEI), namely Assam State for Ranjit rice (RR) and Manipur State for Dharam rice (DR), Black rice (BR), and Thangjim rice (TR) from September through December. Each sampling field zone for a rice variety was made of at least 1 acre, and three rice plant samples were randomly collected from 5 different fields for each of the four varieties once a month. The panicles having false smut grains with black-colored grains in the case of susceptible varieties were observed and collected during November–December. All plants and grains were collected in the early morning before intense sunlight.

### 2.2. Stages in the Life Cycle of the U. virens

Chlamydospores of *U. virens* collected from the infected rice variety Dharam and Thangjim from Manipur state were used for the in vitro studies. The spores were grown in potato dextrose broth (PDB) (HiMedia, Mumbai, India) in triplicates at three different temperatures of 20 °C, 25 °C, and 30 °C in a stationary incubator (ORBITEK, Scigenics Biotech, Chennai, India) and on an orbital shaker incubator (Kuhner SHAKER X, Birsfelden, Switzerland) at 25 rpm. They were also grown in potato dextrose agar (PDA) (HiMedia, Mumbai, India) plates containing chloramphenicol at 25 mg L^−1^ (HiMedia, Mumbai, India). The chlamydospores germinations in PDB were monitored every 6 h for 3 days, while PDA plates were monitored every 24 h for 2 weeks.

### 2.3. Scanning Electron Microscope

To understand how the fungus colonized grain surfaces, a scanning electron microscope (SEM) was used. Briefly, grain surface spikes were removed by scraping them with surgical blades (Ribbel International, Rai, India). The grains of rice varieties BR and DR with and without spikes were dusted with chlamydospores of *U. virens* under dry and wet conditions at 25 °C in sterile Petri plates under a Laminar air flow and were observed under SEM after three days post-inoculation (DPI). SEM-prepared samples were placed on a double-sided adhesive tape mounted on a specimen stub, sputter-coated with gold-palladium, and examined in a Carl Zeiss (Jena, Germany) ΣigmaVP microscope at five kV.

### 2.4. SEM-EDX

The micronutrient composition of the grain surface was measured by using SEM energy-dispersive X-ray spectroscopy (EDX) (Oxford, UK) as previously described [23]. The EDX analysis was carried out on the surface rice varieties. Importantly, the long spikes and the region between the small spikes for all rice varieties (DR, BR, TR, and RR) without any visible fungal infection were analyzed. The measurement was carried out in three triplicates.

### 2.5. Confocal Microscopy

A confocal laser scanning microscope (CLSM) (Leica, Germany) with fluorescent dyes 4′, 6-diamidino-2-phenylindole (DAPI) (Thermo Fisher Scientific, Waltham, MA, USA), wheat germ agglutinin (WGA) (Thermo Fisher Scientific), and propidium iodide (PI) (Thermo Fisher Scientific, Waltham, MA, USA) were used to stain and identify components of the plant and the fungus during the infection [24] of DR grains and also to determine the dead and live cells in the interactome.

### 2.6. Protein Extraction and Quantification

500 mg of each rice variety was taken in separate mortar and pestle and crushed into fine powder by using liquid nitrogen in triplicates. The proteins were extracted using an extraction buffer containing 9 mM CaCl_2_ (Sigma-Aldrich, St. Louis, MO, USA) solution with 5% (*v*/*v*) Triton-X-100 (Sigma-Aldrich, St. Louis, MO, USA), 0.2% (*w*/*v*) polyvinylpyrrolidone (Sigma-Aldrich, St. Louis, MO, USA), and 0.2% (*w*/*v*) dithiothreitol (DDT) (Sigma-Aldrich, St. Louis, MO, USA), 0.2% L-ascorbic acid (Sigma-Aldrich, St. Louis, MO, USA) and a protease inhibitor cocktail (Sigma-Aldrich, St. Louis, MO, USA) and incubated at 4 °C for 1 h. The homogenized sample was transferred to microcentrifuge tubes (Tarsons) of 2 mL and centrifuged for 10 min at 10,000× *g* in a refrigerated centrifuge (Eppendorf 5430R, Hamburg, Germany), and the supernatants were transferred to a fresh tube. A 200 μL of precipitation solution pre-chilled (50 μL of 100% Trichloroacetic acid (Sigma-Aldrich, St. Louis, MO, USA) plus 50 μL of 100% extra pure acetone (Sigma-Aldrich, St. Louis, MO, USA)) was added to 1.5 mL supernatant and incubated overnight at −20 °C in a deep freezer (Eppendorf, Hamburg, Germany) for slow protein precipitation. The proteins were then pelleted at 12,000 rpm for 15 min [25].

The proteins extracted were quantified using Bio-Rad protein assay Kit II (Bio Rad, Hercules, CA, USA), which is based on is based on the Bradford dye-binding method [26]. Three dilutions of the protein and standards of bovine serum albumin (BSA) were made, which were representative of the protein solution. A total of 800 µL of each standard and sample solution was put into a clean microcentrifuge tube in triplicate, and 200 µL of dye reagent concentrate (Bio Rad, Hercules, CA, USA) was added to each of the tubes, vortexed, and incubated at room temperature for 10 min. The absorbance was measured at 595 nm in a spectrophotometer (Eppendorf BioSpectrometer, Hamburg, Germany).

### 2.7. SDS-PAGE

The changes at the proteome level in the plant–fungus interaction were studied using sodium dodecyl-sulfate polyacrylamide gel electrophoresis (SDS-PAGE). For SDS-PAGE, 250 µg/µL protein samples were loaded and run using Mini-PROTEAN Tetra Cell apparatus at 90 V (Bio–Rad, Hercules, CA, USA) in 12% gels with a molecular weight marker (Bench Mark^TM^ Protein ladder, Life Technologies, Bangalore, India). Gels were stained using a solution of 0.2% (*w*/*v*) Coomassie Brilliant Blue R250 (Sigma-Aldrich, St. Louis, MO, USA), methanol (Sigma-Aldrich, St. Louis, MO, USA) 40% (*v*/*v*) and acetic acid (Sigma-Aldrich, St. Louis, MO, USA) 10% (*v*/*v*) for 90 min and destained with a solution of methanol 40% (*v*/*v*) and acetic acid 10% (*v*/*v*) for 90 min and later stored in acetic acid 5% (*v*/*v*) at room temperature.

### 2.8. 2DE SDS-PAGE

For performing a 2D gel electrophoresis SDS-PAGE, a total of 250 µg/µL protein was loaded on 11 cm immobilized pH gradient (IPG) strips (Bio-Rad, Hercules, CA, USA) with a pH gradient of 3–10 (Bio–Rad, Hercules, CA, USA). The strips were rehydrated overnight at room temperature (20 °C) with Ready-Prep rehydration buffer (Bio–Rad USA). Isoelectric focusing was performed in a PROTEAN II xi Cell System apparatus at 180V (Bio–Rad, Hercules, CA, USA). Gels were stained using a solution of 0.2% (*w*/*v*) Coomassie Brilliant Blue R250, methanol 40% (*v*/*v*), and acetic acid 10% (*v*/*v*) for 90 min and destained with a solution of methanol 40% (*v*/*v*) and acetic acid 10% (*v*/*v*) for 90 min and later stored in acetic acid 5% (*v*/*v*) at room temperature.

The gels were calibrated with 2D SDS-PAGE Standards #1610320 Bio-Rad and imaged in Bio-Rad Gel Doc System and analyzed using PDQuest v.8.01 (Bio–Rad, Hercules, CA, USA). Spots were aligned and matched after spot detection and background subtraction (mode: average on the boundary). A quantitative determination of the spot volumes was performed (mode: total spot volume normalization). Triplicate gels were analyzed to ensure reproducibility between normalized spot volumes. Only spots that showed more than 1.7-fold changes in the average volume between segments, with differences indicated as statistically significant by Student’s *t*-test (*p* < 0.05), were defined as differentially expressed protein spots. However, failed 2-D SDS-PAGE spots matching between the replicate gels led us to use bands from the SDS-PAGE gels for LC-MS/MS analysis.

### 2.9. Sample Preparation and LC-MS/MS Analysis

Differentially expressed protein bands were excised manually from the gels. Each excised gel piece containing proteins was destained, and trypsin digestion was performed using the In-Gel Digests (IGD) kit (Product Code PP0100) according to the manufacturer’s protocol (Sigma-Aldrich, St. Louis, MO, USA). After digestion, the protein peptides solution was lyophilized and stored at −80 °C until use. Electrospray ionization (ESI)-based LC-MS/MS analysis was performed according to the protocol used by the Centre for Cellular and Molecular Platforms of the National Centre for Biological Sciences (NCBS), Bangalore, India. The digested peptides were then reconstituted in 15 μL of 2% acetonitrile (ACN) (Sigma-Aldrich, St. Louis, MO, USA) and 0.1% formic acid (Sigma-Aldrich, St. Louis, MO, USA), and 1 μL of the samples was analyzed. The digested peptides were subjected to 70 min reverse-phase liquid chromatography (RPLC) gradient, followed by the acquisition of the data on the Thermo Scientific™ LTQ Orbitrap XL™ Hybrid Ion Trap Orbitrap Mass Spectrometer, and the generated data were searched for the identity of the peptides using SPIDER and MASCOT against all updated entries in the *Oryza_Sativa*_Uniprot 20180813 (168,235 sequences; 59,738,601 residues) and Uniprot_*Ustilaginoidea_virens* 20190304 (13,254 sequences; 6,180,358 residues) databases. For validating the identified peptides, a MOWSE probability threshold score (*p* < 0.05) was set, and a false-discovery rate (FDR <1%) [27] was calculated using the decoy database (http://www.matrixscience.com/help/decoy_help.html, 28 November 2021).

## 3. Results

The rice varieties, especially the susceptible rice grains of Dharam rice (DR) and Thangjim rice (TR) variety contained chlamydospores of *U. virens* over the grains before the visible smut formation (Figure 1 and Figure 2), which was not observed on the rice grains of resistant Black rice (BR) and Ranjit rice (RR) (Figure 2).

Fungal hyphae were transversing the entire region of the cross-sectioned portions of false smut grains of the Dharam variety infected with *U. virens* (Figure 3 and Figure 4). The spikes on the surface of the husk of susceptible grains of DR and TR and that of resistant grains from BR and RR were of different sizes (Figure 5). The shape of the spikes was mainly of two types, the smaller spikes (SS) had a triangular base, and the other spikes were long hair-like (LS) with a circular base. The longest spikes found in BR were 330.76 ± 1.24 µm long and 23.62 ± 0.41 µm in breadth at the center, while the Dharam rice variety, which is susceptible to fungus, had a spike length (LS) of 278.36 ± 0.86 µm and breadth of 26.64 ± 0.19. The long and small spikes frequency per 200 µm^2^ varied between rice varieties, and the density was higher for BR with 3 ± 1.41 for LS and 19.2 ± 1.78 for SS, while for DR rice variety, it was 1.6 ± 0.54 for LS and 9 ± 0 for SS (Appendix A).

The elemental analysis using EDX showed variations on the grain surface of various varieties. Primarily, the carbon (C) content was low in the small spikes (SS) area of the resistant variety (BR and RR) compared to susceptible varieties (DR and TR). The silicon content of susceptible rice TR was very low, with 7.77% ± 0.15 on Long spikes (LS) and 20.20% ± 0.16 in the (SS) region, and it was 30.55% ± 0.14 (LS) and 21.52% ± 0.41 (SS) for DR rice. In comparison, the resistant BR rice contained 23.39% ± 0.22 for LS and 29.31% ± 0.17 for the SS region, and RR rice had 27.83% ± 0.16 for (LS) and 25.77% ± 0.12 for SS (Figure 6; Appendix A).

The *U. virens* spore exines were in different contours, mainly in the form of thorns and hooks (Figure 7). The matured chlamydospores of *U. virens* were of the size 4.82 ± 0.85 µm (dehydrated) under the scanning electron micrograph, while under confocal, they were 6.56 ± 0.13 µm (rehydrated). After germinating on the PDA medium, the fungus forms hyphae and later ascospores and secondary conidial cells capable of infecting susceptible varieties (Appendix A).

Chlamydospores of *U. virens* adhere to both susceptible DR and resistant BR rice varieties with and without spikes but do not grow over them under dry conditions even after 3 DPI (days post-inoculation) (Appendix A). However, under the wet condition, chlamydospores form hyphae on DR variety at 3 DPI with the presence or absence of spikes having no effect (Appendix A). While under wet conditions, chlamydospores form hyphae on the BR variety in the absence of the spikes but have no fungal hyphal growth in the presence of the spike (Appendix A).

The infection rate in terms of the number of grains among the infected plantlets of the susceptible rice variety Dharam rice per infected plant when false smut grains were fully formed was 9.66% ± 19.35, while that of variety Thangjing rice variety was 2.14% ± 0.84, and the resistant Black rice and Ranjit rice grains were never found to be infected with *U. virens* (Appendix A).

Differentially expressed bands on SDS-PAGE were used for LC-MS/MS analysis. One thousand five hundred nineteen peptides were detected from the eight bands after sequencing, and most of them were identified except some, which were uncharacterized (Appendix A). There were 23 peptides detected, which were expressed by the non-infected Dharam rice grains and absent in the infected DR grains. There were 256 peptides detected in band 1, 17 peptides in band 2, 309 peptides in band 3, 332 peptides in band 4, 323 in band 5, only 6 peptides were detected in band 6, while 94 peptides in band 7, and 182 peptides were detected in band 8 (Figure 8 and Figure 9).

## 4. Discussion

The false smut of rice grains in small numbers was recognized by farmers as a sign of plentiful crop yield and seen as an insignificant event. In recent times, the false smut has become epidemic and more severe in farms where high fertilizer-receptive rice hybrids are grown.

This study sought to determine proteome changes in rice grains due to the infection of the fungus *U. virens*. One intriguing question of importance in this research was why *U. virens* do not infect all rice grains of a susceptible variety in a particular habitat and also why some rice varieties, such as Black rice, are always resistant to this infection.

The chlamydospores were found to be present even before the disease symptoms appeared in the susceptible variety and could have acted as primary inoculum in the false smut disease of rice. The fungus *U. virens* has previously been found to infect coleoptiles and the roots of seedlings during the germination stage [28]. Nonetheless, such infections are not systemic as no hyphae are observed below the pedicel and do not form the characteristic symptom of a smut ball [29,30]. Other studies indicated that initial infections might occur in rice pistils [31,32]. In addition, it was shown that *U. virens infect* rice filaments initially before the flower opening [16,33]. Previous studies have shown that the conidia of *U. virens* germinated under proper conditions and produced enormous secondary conidia and hyphae over rice spikelets [34] (Fan et al., 2014) and entered the spikelet’s inner spaces [14,29,35].

Previous studies have found that the compatible interaction of *U. virens and rice* prevented pollen production, ovary fertilization, and the flower-opening process [13,14,31]. Moreover, the infection of *U. virens* downregulates the expression levels of defense-related genes *but* activates the expression of genes associated with grain filling, including the endosperm-specific transcription factors, starch anabolism genes, seed storage protein genes, and fertilization, indicating that *U. virens takes* advantage of rice nutrients supply systems by unknown mechanisms to form rice false smut balls [13,14]. Meanwhile, many pathogenesis-related genes are induced to activate the rice resistance signaling pathways in incompatible interactions [36]. This indicates that the pathogen has evolved pathogenicity to infect a host without halting the host’s ability to produce defense-related proteins that are active against other pathogens.

There have been no reports to the best of our knowledge on the role of spikes on the surface of the grains in their quality or resistance to pathogens. The small spikes areas, also affected while removing the long spikes, had some role to play in the smut formation as the carbon content near the area of the small spikes of TR was 1.63-fold greater than the resistant varieties. On the other hand, silicon content was lower in the susceptible rice, indicating the possibility of high carbon: silicon ratio is a favorable factor for *U. virens* infection, which needs to be further confirmed by using more rice varieties. Si is known to effectively diminish various environmental stresses and enrich plants’ resistance against microbial pathogens by activating defense-related enzyme activities, stimulating antimicrobial compound production, regulating the complex network of signal pathways, and activating the expression of defense-related genes. It can act as a modulator prompting plant defense responses and stress signaling systems leading to induced resistance [37]. It was found that Si in the leaf epidermis confers resistance against appressorial penetration by the rice blast fungus *Magnaporthe oryzae,* and also, the total number of lesions per leaf was inversely proportional to the amount of Si applied [38]. It can be assumed that Si will be a potential candidate for fertilizer input in future rice cultivations.

The growth of fungus on otherwise resistant Black rice variety upon the removal of spikes is one piece of evidence indicating that spikes might be playing a critical role in the resistance to *U. virens* infection. At the microscopic level, we show that chlamydospores, after germination, could infect and grow on the Dharam rice grains under wet conditions with and without the spikes on its surface. No infection was observed on the grains of Black rice with the spikes, although *U. virens* grew over the spike-removed grains after 3 DPI at 25 °C. This finding shed light on the cause of false smut to be multifactorial, with the intrinsic properties (spikes and various proteins) of the rice grain and moisture content as one of the main limiting factors for infection. Furthermore, the inherent nature of the spikes on the resistant BR variety contributed to the resistance to fungal infection.

Not all grains in a panicle and all the sets of panicles or plants of a susceptible variety (Dharam and Thangjim) were infected at a particular period, which could be mainly due to the absence of the fungus inoculum and availability of proper conditions, particularly the moisture and also inherent characteristics of those grains such as the elemental composition and spike lengths. It was found that the spikes over the husk on a grain could be crucial for establishing the fungal conidial interaction, which needs to be further studied with varying intensity of light, temperature, humidity, and wind speed. The other practical application of this research project was to develop markers to screen varieties that are inherently resistant to the fungus and propagate it. In addition, to use protein markers to check the new variety developed by various techniques, such as plant breeding, having the proteins that would be resistant to the smut before being marketed on a large scale (Figure 8 and Figure 9 and Appendix A).

Analysis of global proteins has become the best way to study the expression of various genes involved in an interaction system, particularly for organisms with limited genomic resources. They are mainly carried out by doing a proteome database similarity search and are considered an excellent means of identifying proteins rather than deciphering their respective genes.

The relationship between the rice varieties and the effector diversity of *U. virens* involved in pathogenicity remains unclear and needs to be studied further. Effectors secreted *by U. virens* during infection have been reported to play a significant role in suppressing the host defense and converting the plant’s nutrition for its growth and development [39]. Previously, it was predicted that *U. virens* could discharge 628 peptides, of which 18 peptides could act as effectors suppressing hypersensitive responses, and 13 had the potential to induce plant cell death in *Nicotiana benthamiana* [16].

Effectors may be recognized by resistance or sensitivity receptors in the host, determining disease outcomes.

During false smut of rice formation, the fungus (*U. virens*) and the rice (DRS) were producing superoxide dismutase (SOD), inferring the molecular war going on between them. The interaction between them can be considered as a battleground where the fungus has been armed with SODs designed to overcome the plant’s defense in the aggressive microenvironment of the host–pathogen crossing point [40]. It has been observed that the superoxide dismutase (SODs) within a cell organize the first line of defense against reactive oxygen species (ROS) [41]. The SODs are also considered to be decisive enzymes essential to sustain the redox potential of the cells. It plays a vibrant role in shielding regular cells from reactive oxygen species (ROS) formed throughout numerous intracellular pathogen infections [42].

During smut formation, *U. virens* was producing small secreted protein (SSP) coded by genes, which lack recognized domain or resemblances to identified protein sequences. They have been reported to suppress the host resistance response and alter its working. An effector such as SSP1 from the white rot fungus *Pleurotus ostreantus* has been found to potentially regulate the ligninolytic systems, secondary metabolism, development, and fruiting body [43].

The proteins expressed by the non-infected Dharam grains were not expressed in infected Dharam rice grains either due to its degradation by the *U. virens* infection process or due to suppression by the effectors of fungus. The low molecular weight proteins in band 8 contained proteins, several of which were uncharacterized and could have played unknown virulence roles during the infection process. Resistant Black rice expressed 1-Cys-peroxiredoxin and prolamin peptides located in band 1 have been reported to be involved in cell redox homeostasis. These peptide bands were not prominent in other *U. virens*–rice grain interactions suggesting the peptides may play a role in mitigating the fungus growth in Black rice. We observed the expression of 1-Cys peroxiredoxin, allergen RA5B, oleosin, and Bowman–Birk-type bran trypsin inhibitor in non-infected susceptible Dharam rice. These proteins have the potential to inhibit endopeptidase activity that *U. virens* employ to colonize its host. From infected grains, we identified a serine endopeptidase activity peptide (located at band 7, Figure 8) that catalyzes the cleavage of peptide bonds in proteins. This evidence that *U. virens* expresses protein hydrolytic enzymes and justifies the absence of specific peptide bands in infected grains (Figure 8). Other peptides that clustered in band 7 showing enzymatic activities included 1,3-β-glucosyltransferase and glucan endo-1, 3-β-D-glucosidase, which are plant cell wall degrading enzymes critical for pathogenicity.

## 5. Conclusions

The microscopic and proteomic investigation revealed that the spikes, their elemental composition along with rice proteins and the moisture present on the surface of the grain varieties are critical in the formation of smut, of which the major contribution to resistance is from the intrinsic (genetic) nature of the grain. The proteins cysperoxiredoxis, prolamin, 1-Cys peroxiredoxin, allergen RA5B, and oleosin, with other uncharacterized proteins, could be critical in preventing the growth of *U. virens* on grains that remained uninfected. The uncharacterized proteins that could play a specific role of effectors and enhanced host specificity need further investigation.

## Figures and Tables

**Figure 1 biomolecules-13-00669-f001:**
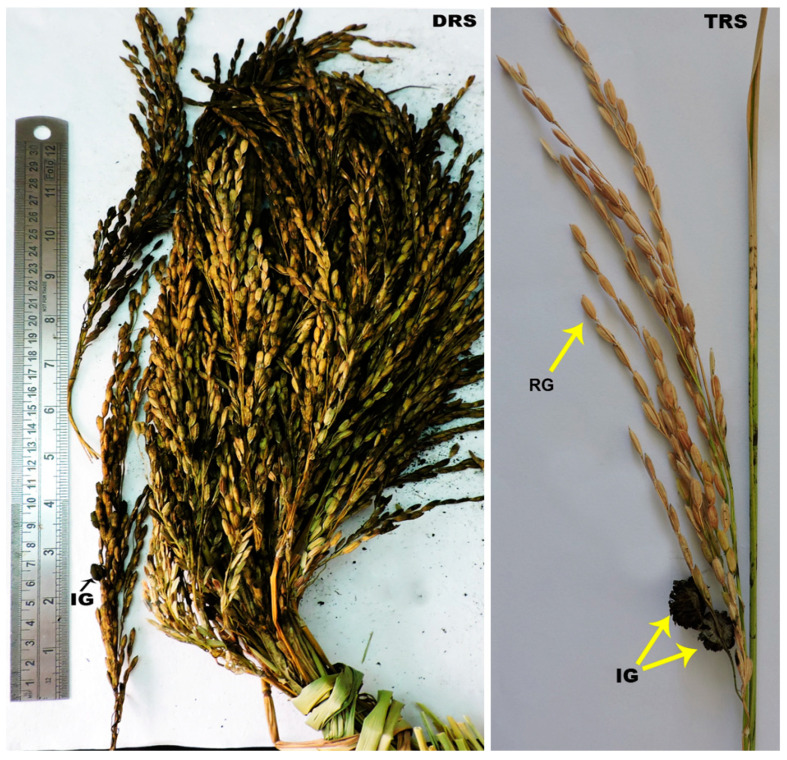
Panicle from infected rice varieties Dharam (DRS) and Thangjim (TRS) susceptible to *U. virens* portraying false smut of rice. IG is infected grain and RG is rice grain uninfected. The scale bar is 30 cm.

**Figure 2 biomolecules-13-00669-f002:**
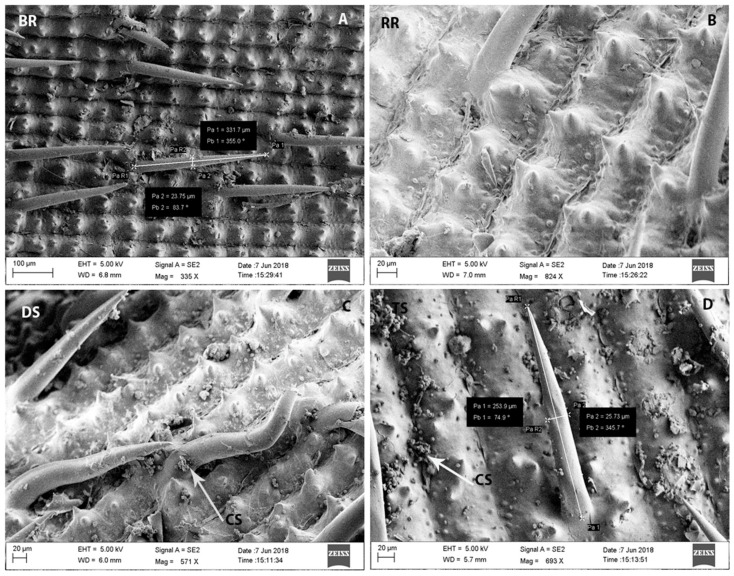
SEM micrographs for varieties of rice grain surfaces: (**A**,**B**) Resistant varieties (BR and RR) have spikes of different sizes. (**C**) Susceptible Dharam rice (DS). (**D**) Susceptible Thangjim rice (TS). Varieties (DR/DS and TR/TS) are infected with *U. virens* bearing chlamydospores (CS/C). The scale bar for (**A**) = 100 μm and (**B**–**D**) = 20 μm. Note: Pa 1, Pa 2 (Length of the spike, and the angle respectively), Pb 1, Pb 2 (Length of the spike, and the angle respectively), Pa R1, and Pa R2 (The lines get named by default in the SEM machine).

**Figure 3 biomolecules-13-00669-f003:**
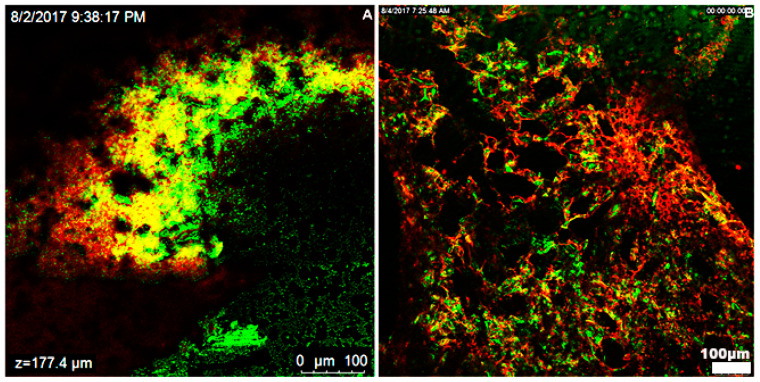
Confocal microscopical view of *U. virens* interaction with the host using WGA and PI fluorescent stains. (**A**,**B**), The chlamydospores are green to black, and plant cells appear red, while the fungal hyphae and their secretions are yellow; bar; (**A**,**B**) = 100 µm. Note: z is the thickness of the sample.

**Figure 4 biomolecules-13-00669-f004:**
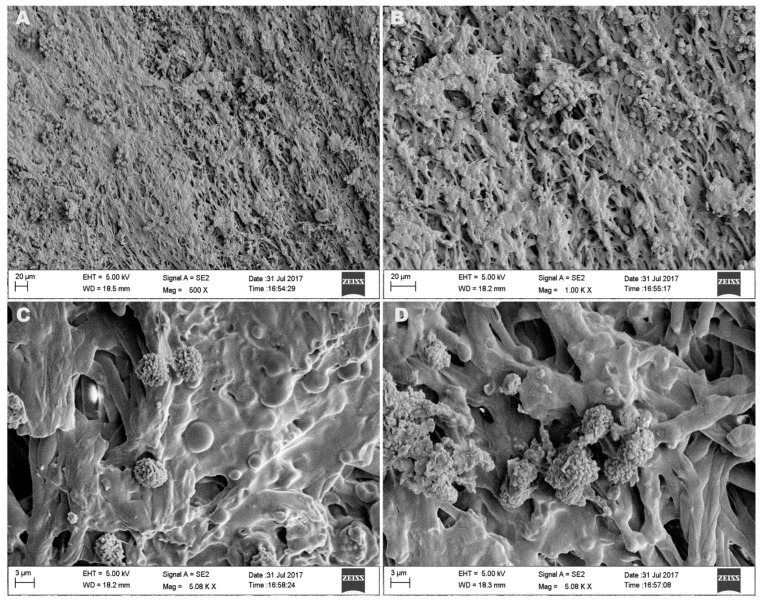
Scanning electron micrographs of the transverse, longitudinal cut sections of infected Dharam rice. (**A**,**B**), The chlamydospores of the fungal pathogen *U. virens*, along with the hyphae, spread throughout the interior of the infected grain; bar (**A**,**B**) = 20 µm; (**C**,**D**) also at higher magnification; bar (**C**,**D**) = 3 µm.

**Figure 5 biomolecules-13-00669-f005:**
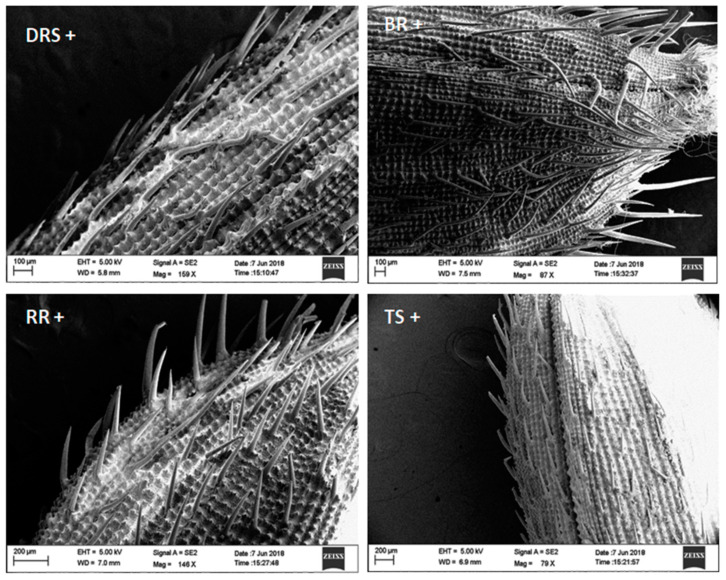
SEM micrographs for rice grains depicting different spike-like hair orientations for Dharam rice (DRS+, susceptible), Black rice (BR+, resistant), Ranjit Rice (RR+, resistant), and Thangjing rice (TS+, susceptible).

**Figure 6 biomolecules-13-00669-f006:**
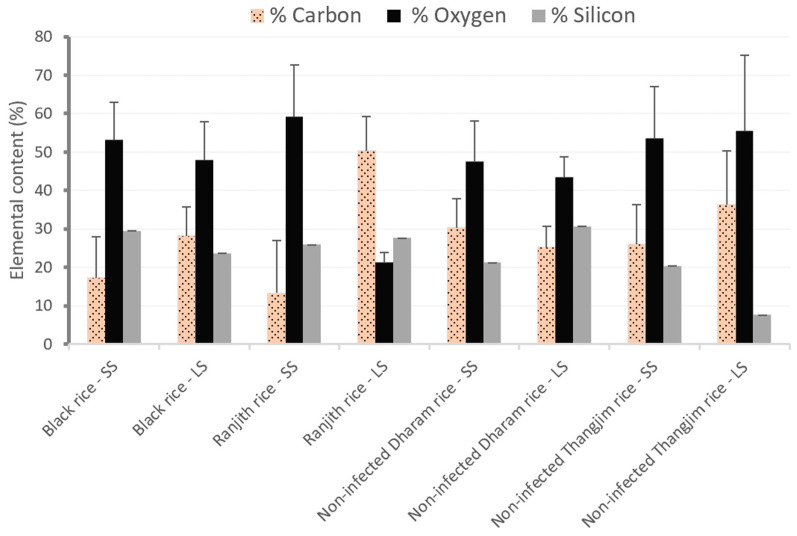
Relative distribution of carbon, oxygen, and silicon in long spikes (LS) and small spikes (SS) region in different rice varieties. Bar represents standard errors.

**Figure 7 biomolecules-13-00669-f007:**
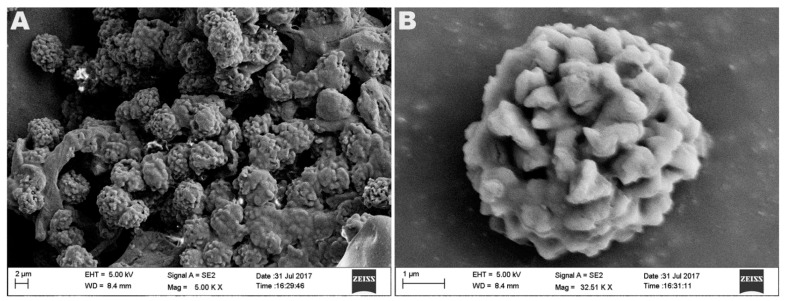
Scanning electron micrographs of *U. virens* chlamydospores. (**A**,**B**), Chlamydospores of the fungal pathogen *U. virens* on the surface of the Dharam rice variety at various magnifications; bar (**A**) = 2 µm, (**B**) = 1 µm.

**Figure 8 biomolecules-13-00669-f008:**
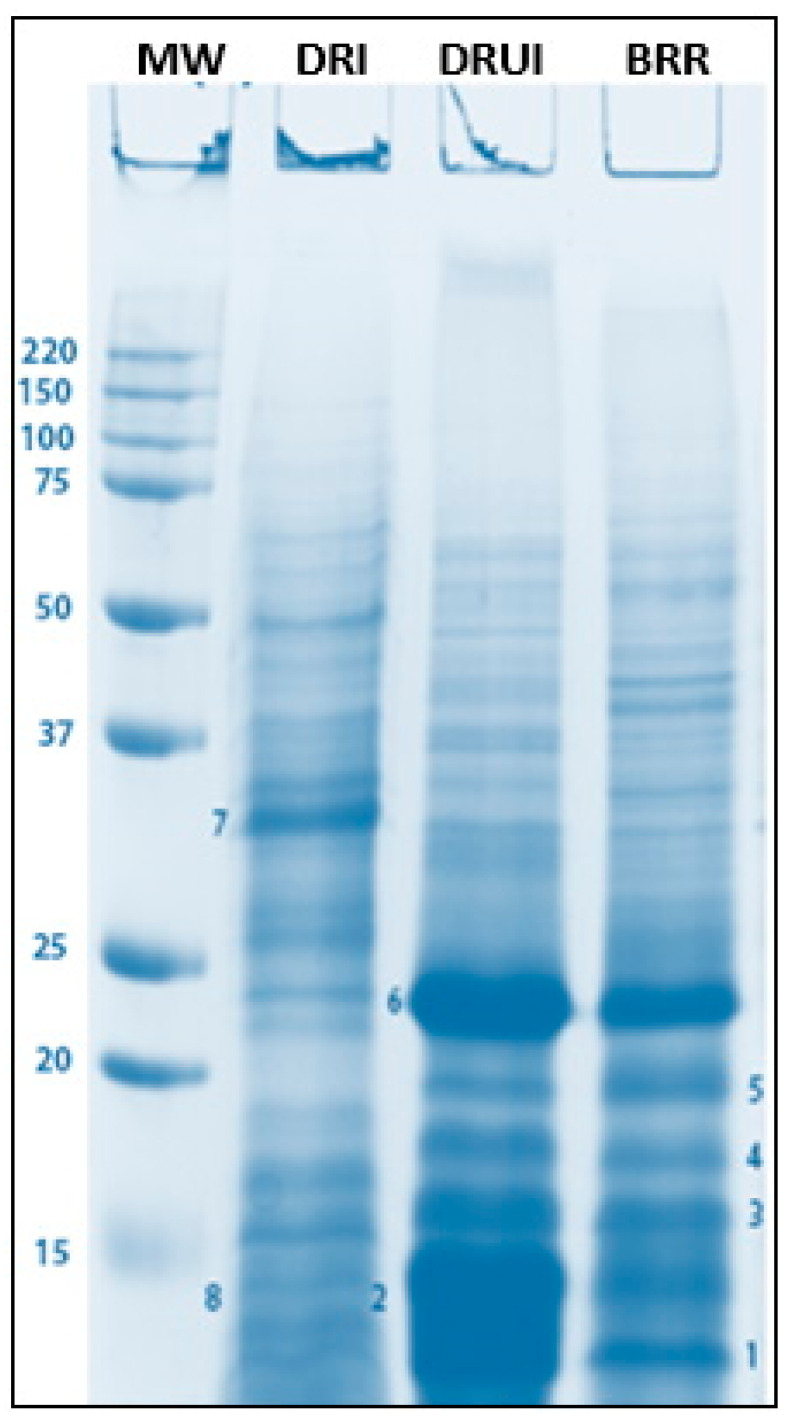
SDS-PAGE profile of the protein samples taken from the infected and non-infected rice grains of the Dharam variety and the resistant Black rice variety. Note: MW = molecular weight marker, DRI = Dharam rice grains infected, DRUI = Dharam rice grains non-infected, and BRR = Black rice-resistant variety.

**Figure 9 biomolecules-13-00669-f009:**
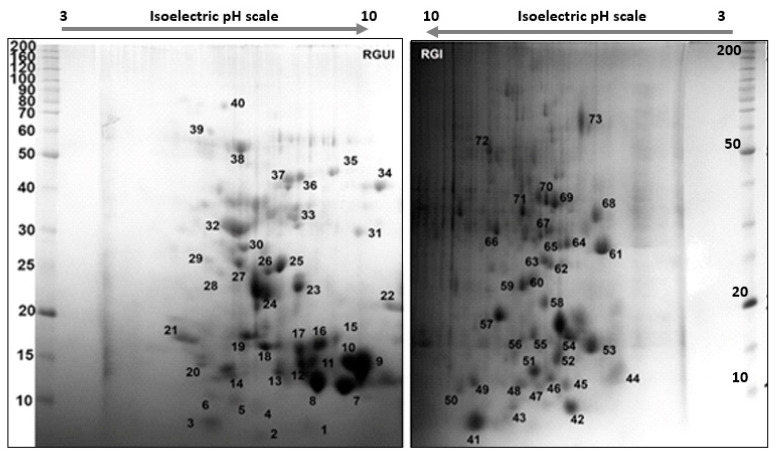
Two-dimensional gel electrophoresis image of the protein samples taken from the non-infected and infected rice grains (of Dharam variety), respectively, after IEF in 3–10 IPG strips. Note: RGUI = rice grain non-infected, RGI = rice grain infected. Note: 1–73 are high to moderately intense protein spots which are easily visible in white light).

## Data Availability

Data is contained within the article or Appendix A.

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
