# Peer review of "Grain Characteristics, Moisture, and Specific Peptides Produced by Ustilaginoidea virens Contribute to False Smut Disease in Rice (Oryza sativa L.)"

_biomolecules, 2023, doi:10.3390/biom13040669_

Round 1
Reviewer 1 Report
In the current manuscript, microscopic and proteomic analyses were performed by comparing Ustilaginoidea virens infected and non-infected rice of the susceptible and resistant rice varieties to provide insights into the molecular and ultrastructural factors involved in false-smut formation. It includes a molecular analysis of the proteins involved in a plant-fungus interaction. The manuscript is well-written and reports a novel aspect of a biological event. However, morphological and molecular identification of the fungus is missing. Additionally, microscopic slides are required for the comparative study. The minor points have been provided in the manuscript file (attached) and below:
L19-20: For the first time, the full name of the terms must be written, and the abbreviated term must be in parentheses.
-In the introduction, the amount of global rice production and the amount of its production in India should be mentioned.
L79 & 81: Write the name of the company producing the media cultures.

Author Response
We completely agree with the reviewer’s comments and give thanks for the valuable suggestions and the manuscript has been modified accordingly.
Reviewer 1 Comments
The manuscript shows interesting new information concerning the proteome of Ustilago esculenta, however before be published there are some suggestion that should be accepted. In the current manuscript, microscopic and proteomic analyses were performed by comparing Ustilaginoidea virens infected and non-infected rice of the susceptible and resistant rice varieties to provide insights into the molecular and ultrastructural factors involved in false-smut formation. It includes a molecular analysis of the proteins involved in a plant-fungus interaction. The manuscript is well-written and reports a novel aspect of a biological event. However, morphological and molecular identification of the fungus is missing. Additionally, microscopic slides are required for the comparative study. The minor points have been provided in the manuscript file (attached) and below:
Author's response:
Thank you for the valuable comments and suggestions which have greatly helped us to improve the manuscript. For morphological characterization, we have observed them under various microscopes including SEM. we could clearly identify the features which are unique to U. virens moreover we have obtained very high scores for the proteins from the fungus matching to U. virens in the UniProt database (an additional raw excel file is included). We have also included the citations which mention that false smut of the rice is characterized by the formation of easily identifiable grains in the form of black balls. We have added all the additional microscopic slides used for the comparative study, please check the supplementary file.
L19-20: For the first time, the full name of the terms must be written, and the abbreviated term must be in parentheses.
Author's response:
We have made sure that all the terms are written in full, and their abbreviated terms are written in parentheses.
In the introduction, the amount of global rice production and the amount of its production in India should be mentioned.
Author response:
As suggested by the reviewer we have mentioned the global rice production and the amount of contribution by major producers including India
L79 & 81: Write the name of the company producing the media cultures.
Author response:
We have included the brand/ company names of all the media cultures, chemicals, and instruments used for the research work.
Reviewer 2 Report
Dear Authors,
I have some suggestions to improve their work.
The manuscript submitted needs to be entirely in the format suggested by MDPI. For example, the number section needs to be included; the references do not have the journal format.
In the References section, there is no number for each article.
In the text, the citation of the articles does not correspond to what was requested by the journal.
The references are not in the format and order suggested.
The Author Contribution, Funding, Institutional Review Board Statement, Informed Consent Statement, Data Availability Statement and Conflicts of Interest sections need to be included.
A grammatical revision is required throughout the text; for example, some articles, commas, adverbs, etc., must be included.
The manuscript needs to screen for typos and grammatical errors. Keep the same font style and size throughout the manuscript.
Some significant points should be addressed before the manuscript can be considered for publication.
The methods need more details of the experiments, are poorly described, and reagents and equipment used, such as brands, for example. Some references are missing, for example, in Stages in the Life cycle of the U. virens section and microscopy sections.
The protein extraction and quantification protocol need to be included.
The discussion is poor; the authors rephrase the results. However, there is no genuine discussion about the dimension of rice grain spikes, their elemental composition, moisture content, and proteins produced by the grains and the fungi U. virens, which play a vital role in the formation of false smut.
Author Response
We completely agree with the reviewer’s comments and give thanks for the valuable suggestions and the manuscript has been modified accordingly.
Reviewer 2 Comments
The manuscript submitted needs to be entirely in the format suggested by DPI. For example, the number section needs to be included; the references do not have the journal format. In the References section, there is no number for each article. In the text, the citation of the articles does not correspond to what was requested by the journal. The references are not in the format and order suggested. The Author Contribution, Funding, Institutional Review Board Statement, Informed Consent Statement, Data Availability Statement, and Conflicts of Interest sections need to be included.
Author response:
The current version of the manuscript is entirely in the format suggested by DPI.
A grammatical revision is required throughout the text; for example, some articles, commas, adverbs, etc., must be included.
Author response:
A grammatical revision has been done throughout the text; using the paid version of
Grammarly and checked by a native speaker.
The manuscript needs to screen for typos and grammatical errors. Keep the same font style and size throughout the manuscript.
Author response:
We have made sure to the best of our abilities to keep the manuscript free of typos and to maintain uniformity in the font style and size throughout the manuscript as per the requirement of the Journal.
Some significant points should be addressed before the manuscript can be considered for publication. The methods need more details of the experiments, are poorly described, and reagents and equipment used, such as brands, for example. Some references are missing, for example, in Stages in the Life cycle of the U. virens section and microscopy sections. The protein extraction and quantification protocol need to be included.
Author response:
We completely agree with the reviewer and have made sure that all the methods used in the research work are explained and cited appropriately.
There is no genuine discussion about the dimension of rice grain spikes, their elemental composition, moisture content, and proteins produced by the grains and the fungi U. virens, which play a vital role in the formation of false smut.
Author response:
The discussion has been improved as per the suggestions of the reviewer.
Reviewer 3 Report
Dear authors,
since false-smut is a very significant disease for rice cultivation and it would pave the way for protective measures the most important way of its management, the subject was found to be prominent. The objective of the paper was found fantastic. The writing of the manuscript is simple and clear. The results of the study are also clear. However, although the aim of the study was pointed out very well, why this study is needed should be explained in a few sentences. In addition, there is no literature statement in the methods except SEM-EDX (2.3.2.).
Mostly the studies that need to be done in the future were given in conclusion, whereas the contribution of the findings obtained in this study to science should be explained fully and clearly in a few sentences.
In addition to these comments, I have some remarks as follows:
In L82, “The chlamydospores germinations in PDB was were monitored every 6 hours….”
“Was” should be changed to with “were”.
In L99, “….and RR) without any visible fungal infection was were analyzed.”
“Was” should be changed to with “were”.
In L183-184, “After germinating on the PDA medium, the fungus form forms hyphae….”
“s” should be added next to “form”
Author Response
We completely agree with the reviewer’s comments and give thanks for the valuable suggestions and the manuscript has been modified accordingly.
Reviewer 3 comments
Since false-smut is a very significant disease for rice cultivation and it would pave the way for protective measures the most important way of its management, the subject was found to be prominent. The objective of the paper was found fantastic. The writing of the manuscript is simple and clear. The results of the study are also clear. However, although the aim of the study was pointed out very well, why this study is needed should be explained in a few sentences. In addition, there is no literature statement in the methods except SEM-EDX (2.3.2.).
Author response:
We thank the reviewer for the positive comments about the work.
As per the reviewer’s suggestion, we have included why this study was needed in the introduction and all the necessary citations for the methods section.
Mostly the studies that need to be done in the future were given in the conclusion, whereas the contribution of the findings obtained in this study to science should be explained fully and clearly in a few sentences.
Author response:
We agree with the reviewer’s comment and the conclusion has been modified which explains the finding clearly in a few sentences.
In L82, The chlamydospores germinations in PDB was were monitored every 6 hours...."
"Was" should be changed to with "were".
Author response:
It has been changed
In L99, M....and RR) without any visible fungal infection was were analyzed:
"Was·should be changed to with "were".
Author response:
It has been changed
In L183-184, "After germinating on the PDA medium, the fungus form forms hyphae...."
s" should be added next to "form"
Author response:
It has been changed
Round 2
Reviewer 2 Report
Dear authors,
All suggestions were adequately addressed. Thank you.
There are some writing details, such as repeated words, but the changes you must make are minimal.
Please wait for the editor's comments and the publisher's indications for the acceptance and publication of the article.